# Analysis of the swine movement network in Mexico: A perspective for disease prevention and control

**Alejandro Zaldivar-Gomez**[1], **Jose Pablo Gomez-Vazquez**[2], **Beatriz Martínez-López**[2], **Gerardo Suzán**[1], **Oscar Rico-Chávez**[1]*

1 Laboratorio de Ecología de Enfermedades y Una Salud, Departamento de Etología, Fauna Silvestre y Animales de Laboratorio, Facultad de Medicina Veterinaria y Zootecnia, Universidad Nacional Autónoma de México, Ciudad de México, México, 2 Center for Animal Disease Modeling and Surveillance (CADMS), Department of Medicine and Epidemiology, School of Veterinary Medicine, University of California-Davis, Davis, California, United States of America

* orich@unam.mx

**Data Availability Statement:** The data underlying the results presented in the study are available from Figshare repository (https://doi.org/10.6084/m9.figshare.c.7253860.v2).

## Abstract

Pig farming in Mexico is critical to the economy and food supply. Mexico has achieved advancements in swine health and established an electronic database that records swine movements (Sistema Nacional de Avisos de Movilización, SNAM). In this study, we characterized swine movement patterns in México between 2017 and 2019 to identify specific areas and periods that require concentrated efforts for effective epidemiological surveillance and disease control. We employed a Social Network Analysis (SNA) methodology to comprehensively describe and analyze the intricate patterns of pig movement. In addition, we sought to integrate swine population density into the analysis. We used metrics to characterize the network structure and identify the most critical nodes in the movement network. Cohesion metrics were used to identify commercial communities characterized by a high level of interconnectivity in swine movements between groups of nodes. Of a cumulative count of 662,255 pig shipments, 95.9% were attributed to slaughterhouse shipments. We observed that 54% of all Mexican municipalities were part of the network; however, the density of the movement network was less than 0.14%. We identified four Swine Production Centers in Mexico with high interconnectivity in the movement network. We detected moderate positive correlations ($\rho \geq 0.4$ and $<0.6$, $p < 0.001$) between node metrics and swine population indicators, whereas the number of commercial swine facilities showed weak correlations with the node metrics. We identified six large, geographically clustered commercial communities that aligned with the Swine Production Centers. This study provides a comprehensive overview of swine movement patterns in Mexico and their close association with swine production centers, which play a dual role as producers and traders within the swine industry of Mexico. Our research offers valuable insights for policymakers in developing disease prevention and control strategies.

**Funding:** The first author, Alejandro Zaldivar-Gomez was supported by the Programa de Becas para Estudios de Posgrado, approved by the Consejo Nacional de Ciencia y Tecnología (CONACYT) (Grant number: 750549). There was no further external funding provided for this study.

**Competing interests:** The authors have declared that no competing interests exist.

## Introduction

In recent years, the swine industry in Mexico has undergone significant expansion, emerging as a critical sector within the economy. This sector is an essential component of Mexican meat production, accounting for approximately 7% of the total production [1]. This makes it the third most important contributor to the country's meat production, after the poultry and beef sectors [2]. From 2012 to 2020, the number of swine in Mexico increased from 15.9 to 18.8 million head, indicating a growth of 18.5% [3]. The rise in the pork business can be attributed to several causes, such as the rising demand for pork, pig farm technology upgrades, and more efficient production procedures [4–8].

In Mexico, pig farms are classified into three different production systems based on the use of technology and infrastructures: *i)* technified, *ii)* semi-technified, and *iii)* backyard [7, 9, 10]. The technified production system encompasses 40–50% of the pig population in Mexico which is estimated at 18.4 million animals and contributes to 75% of the annual pork production (1,200 tons) [11]. Technologically advanced farms in Mexico have achieved production parameters comparable to those of nations such as the United States and Canada, which rank third and seventh in global pork production, respectively [12–14]. Moreover, the sanitary status of the pig sector in Mexico has made significant progress, with the eradication of Classical Swine Fever (CSF) and Aujeszky's disease in 2015 [15].

Despite the success achieved by the pig sector in Mexico, there are challenges related to biosecurity and the control of endemic diseases, such as Porcine Reproductive and Respiratory Syndrome virus (PRRSV) or the Porcine Circovirus (PCV) [16, 17]. Moreover, this sector has additional sanitary challenges due to the risk of introducing the African Swine Fever virus (ASFV) in the Americas [18]. Outbreak of those diseases can have significant economic effects, leading to the loss of animals, decreased production and exports, and the imposition of trade restrictions by other countries [19, 20].

The movement of pigs between farms or the presence of fomites in the vehicles used to transport swine can potentially spread endemic or emerging diseases [21–24]. The analysis of swine movement and its close relationship with production sites play a key role in the promotion of animal health and disease control in the swine industry [25]. Animal movement control has been used to monitor the movement of pigs from origin to destination [26, 27]. This strategy promotes disease detection and control and supports the implementation of compartments, as was recently recommended in countries affected by ASF [28].

In Mexico, the movement of animals is regulated by federal Veterinary Services (Servicio Nacional de Sanidad, Inocuidad y Calidad Agroalimentaria, SENASICA). This regulation includes the individual identification of animals, the supervision of sanitary requirements during transport, and the registration of shipments [29]. In the case of swine movements in Mexico, all shipments are registered in an electronic database (Sistema Nacional de Avisos de Movilización, SNAM) [30]. This platform allows for a nationwide centralized and standardized registry of pig movements, generating valuable data to analyze pig movement patterns. Analyzing the pig movement patterns in Mexico will facilitate evidence-based decision-making regarding health-related concerns and promote the sustainable development of the Mexican swine industry.

Social Network Analysis (SNA) has demonstrated its usefulness in characterizing pig movement patterns and assessing their influence on the spread of diseases [31–34]. In this conceptual framework, farms and other facilities are represented as nodes, and the movements of pigs between these nodes are represented as edges. SNA thus facilitates understanding of the interactions between nodes such as farms, slaughterhouses, and livestock fairs, through movement data. SNA has been effectively used in the analysis of pig movements in Europe [35–37], the

United States of America [38, 39], and recently in some countries in South America, such as Uruguay [40], Ecuador [41] and Argentina [42]. In addition, SNA has proven to be a useful tool for identifying commercial communities formed by a group of farms or other types of sites. These communities are characterized by strong interactions due to more frequent trade of animals among them compared to the rest of the network [43]. Another application of SNA is to assist in the parameterization of transmission models to assess the risk of disease spread [44, 45].

Despite advances in animal identification and the digitalization of movement records in Mexico, these data have yet to be used for the analysis of livestock movement patterns. To date, research has mostly been focused on analyzing a single link of the supply chain in Mexico, namely, from slaughter centers to the final consumer [46, 47]. However, to study mobilization between farms or to marketing centers, it is essential to broaden this approach.

In this study, we looked at pig movement patterns at the municipal level in Mexico from 2017 to 2019. We defined municipalities as the third level of political administration, below the state and national levels. We sought to understand the integration of these high-density areas in the commercialization chain, providing information to improve the animal health policies within the swine industry in Mexico. Descriptive statistical and SNA techniques were applied to analyze the spatiotemporal dynamics of pig mobilization and identify the periods of the year with the highest frequency of movements and the areas where pig movements (both outgoing and incoming) are concentrated. We characterized the relationships of the movements, identifying links and critical nodes and the commercial communities present in the movement network. Finally, we compared the results of the node metrics with swine population and conventionally used swine facility indicators.

## Materials and methods

### Data source

The primary data sources employed in this study were official records obtained from the SNAM, which contained comprehensive information on pig movements in Mexico from 2017 to 2019. The SNAM collects detailed information on each pig shipment, including the origin and destination of the pigs.

The swine population distribution data were obtained from a National Livestock Register (Padron Ganadero Nacional, PGN) [48]. This database is an official record at the municipal level of livestock farms in Mexico and the livestock service providers that support these activities. We also collected information on the distribution of swine marketing centers in Mexico, including livestock fairs registered in the PGN and the pig slaughterhouses, published by SENASICA [49, 50]. All datasets were obtained with updates through 2019. This ensures that the analyses are performed on a similar time scale.

No sensitive or private information was collected from any person during this study. Data was collected in accordance with the regulations of the Mexican government regarding confidentiality and the protection of personal data. To protect the privacy of the information of swine producers, movements were tabulated at the municipality level rather than at the farm level. However, this decision limited the capacity to track individual animals or the farm of origin and destination of each shipment.

### Database construction

The three datasets were geocoded using an official catalog of municipalities published by the National Statistical Office (Instituto Nacional de Estadística y Geografia, INEGI) [51]. Each record in the database corresponds to a movement of pigs and contains information about the

municipality of origin and destination, the registration date, the purpose of the movement, and the number of heads shipped. We detected some movements with apparently extreme values, including shipments containing over 500 pigs. These records could represent multiple shipments; however, they were kept in the analysis as originally entered to maintain the representativeness of the movements.

The purpose of movement is classified into four categories, corresponding to the main stages of swine production in Mexico. These are: *i) slaughterhouse*, related to the shipment of swine for slaughter and processing; *ii) fattening*, which includes the shipments of pigs to specialized facilities where animals are fattened to finishing; *iii) breeding*, for reproductive purposes, such as the sale of breeding sows or boars; and *iv) livestock fairs*, to commercial events or livestock exhibitions.

## Statistical analysis of the movements and distribution of swine in Mexico

We applied an exploratory analysis to describe the number of shipments and pigs involved. The weekly frequency of movements and the average shipment size were analyzed to describe the temporal trend, which was evaluated using the Mann-Kendall Trend Test [52]. We examined the seasonality of shipments by the monthly frequency and the purpose of movements. A one-factor analysis of variance (ANOVA) was conducted to assess potential variations in shipment sizes based on the purpose of movement. Subsequently, post-hoc pairwise comparisons were performed using the Tukey method to identify and compare the differences among group means. All statistical analyses were performed in R version 4.1.2. [53].

We used a hotspot analysis to identify areas with high swine density in Mexico. Hotspot analysis is a spatial analysis and mapping technique that aims to identify spatial clustering [54]. The *Global* and *Local Moran's Index* (*I*) was used to assess the degree of spatial autocorrelation of swine density at the municipality level and detect hotspots. The degree of spatial autocorrelation in the dataset is indicated by *Moran's I* values, which range from -1 to 1 [55]. A value of *I* close to 1 indicates that locations with similar values tend to be close to each other. Conversely, values approaching -1 indicate high dispersion, while values near 0 suggest random distribution. Additionally, we created a Kernel density map to assess the spatial pattern of swine density, with the purpose of comparing it to swine movements [56]. The hotspot and Kernel density maps were constructed using QGIS software version 3.24.0-Tisler [57].

## Network analysis of swine movements

Directed networks were used to analyze the movements of pigs. The municipalities were considered nodes in these networks, while pig movements were represented as connections or edges between municipalities. These edges were weighted by the number of shipments between pairs of nodes. The complete network of swine movements was fragmented into subnetworks according to the purpose of movement, allowing for a more detailed and focused analysis of pig movement patterns within specific production processes. To analyze the stability of the pig movement network during the analysis period, the complete network was divided into weekly intervals.

All networks of swine movements were described using the number of nodes (municipalities) and edges (movements). The number of shipments between nodes, the average size of shipments and the Euclidean distance between nodes were calculated to describe the networks. Euclidean distance between pairs of nodes was calculated from the centroids of each municipality.

We calculated metrics to evaluate the structure and connectivity of all networks of swine movements. These metrics have been described in the scientific literature and used to analyze

livestock movement patterns [31, 58–60]. Density and diameter metrics were used to evaluate the degree of connectivity and size of all networks of swine movements. Network density is a metric that quantifies the level of interconnection between nodes relative to the total number of potential connections. The diameter quantifies the extent of indirect paths between pig movements, as it represents the maximum length of the shortest path between nodes.

The reciprocity, transitivity, and assortativity metrics were used to analyze the interaction of pig movements between nodes. Reciprocity assesses the degree of symmetry of the flow of shipments between nodes by calculating the proportion of connections with mutual shipments. Meanwhile, transitivity quantifies the tendency for nodes to cluster or establish tightly interconnected groups within the network. Finally, the assortativity coefficient was used to evaluate the similarity between the connected nodes according to the size of their swine populations. Hence, a positive assortativity coefficient implies a network bias towards a prevalence of connections among municipalities with similar swine populations, while a negative coefficient indicates differing swine populations among connected municipalities.

The impact of nodes in all networks of swine movement was evaluated using degree centrality metrics. A high in-degree indicates that a node receives pig shipments from multiple nodes, while a high out-degree indicates that the node sends shipments to other nodes. Also, the influence of municipalities as hubs and authorities was evaluated. The identification of hubs and authorities is carried out by considering the out-degree and in-degree of a municipality, as well as the number of connections it has to other municipalities with high connectivity. Betweenness was used to quantify the frequency at which a particular municipality acts as an intermediary between pairs of other nodes. The *igraph* package [61] was employed for the construction and analysis of all networks of swine movements.

The Spearman rank correlation coefficient was employed to assess the relationship between the results of node metrics and the variables associated with swine population and commercial facilities. The variables related to the swine population were pig density, the number of farms (categorized into technified, semi-technified, and backyard farms), and the number of technified farms per municipality. Swine commercial facility variables included the number of livestock fairs and pig slaughtering establishments per municipality.

### Detection of commercial communities

The *Walktrap* algorithm was used to identify commercial communities in the complete network of swine movements [62]. These communities are densely connected nodes, defined by pig shipments [63]. The algorithm is efficient in identifying large networks and uses short random walks between nodes to calculate distances between nodes [64, 65]. A weighted random walk selection process was applied to identify communities with the strongest connections, based on the number of shipments per connection. A threshold of 20 or more nodes was set for a large community to be considered. Below this threshold, communities with 3 or fewer nodes were found. In addition, the number of internal and external connections between communities was compared.

## Results

### Descriptive analysis of swine movements

In the 2017–2019 study period, a total of 662,255 pig shipments and 92,224,219 pigs shipped were registered in the SNAM. The most frequent recorded purpose of pig shipments was to slaughterhouses, accounting for 95.9% of movements, 3.1% of movements were for fattening, 0.90% for breeding, and 0.03% for livestock fairs. The average size of the shipments was 139 pigs; the 50th, 75th, and 95th percentiles were 140, 210, and 250 pigs, respectively; the maximum

**Table 1. Size of pig shipments in Mexico by purpose of movement.**

| Purpose of movement | Head per shipment | | | ANOVA[a] | |
|---|---|---|---|---|---|
| | Mean (range) | SD | Median | *F* | *p* |
| Slaughterhouse | 126.5 (1–5,000) | 92.1 | 132 | 75,718 | *< 0.001* |
| Fattening | 550.4 (1–4,517) | 496.1 | 400 | | |
| Livestock fair | 22.7 (1–600) | 67 | 5 | | |
| Breeding | 74.4 (1–1,600) | 123.8 | 15 | | |

[a] The mean differences calculated by Tukey method show that the four groups differ from each other.

number of head mobilized was 5,000 (Table 1). The ANOVA results revealed statistically significant differences with in at least one of the groups based on the purpose of movement. All combinations of the analyzed groups showed statistically significant differences between them, as determined by the Tukey method.

A gradual, but steady increase in the weekly frequency of pig movements was evident over the study period without complete stabilization (Fig 1A), which was confirmed by significant positive result of the Mann-Kendall Trend Test ($\tau = 0.492$, P < 0.001). Similarly, a gradual increase in the average size of shipments was detected, with significant fluctuations (Fig 1B). A seasonal variation in the frequency of pig movements was observed. The highest peaks of shipments were observed from September to December, while January to April had the lowest levels. A significant reduction of 24,373 shipments was observed during this period, which represents a decrease of 10.5% compared to the peak. This seasonal trend was evident in pig movements for fattening and slaughterhouses (Fig 1C).

Shipments for fattening showed the most growth in terms of average shipment size between 2017 to 2019, with a 34% increase from 372 to 702 pigs per shipment. In comparison, these shipments were almost four times larger than those directed to slaughterhouses. Shipments for breeding showed a moderate increase, from 67 to 80 pigs per shipment. The least growth in shipment size was for slaughter, from 123 to 130 pigs per shipment. Significant differences in shipment size were identified through ANOVA analysis, and the Tukey test further indicated that all pairwise comparisons were statistically significant, except for movements for livestock fairs.

## Geographical distribution of pig farms and swine movements in Mexico

The analysis of the geographic distribution of swine movements revealed a concentration of incoming and outgoing shipments in the central region of Mexico, with several important regions in the periphery of the country (Fig 2A). The spatial patterns of these movements are closely linked to the areas of highest swine density in Mexico (Fig 2B and 2C).

The Global Moran's Index revealed spatial autocorrelation ($I = 0.229$, P < 0.05) in swine density in Mexico. Four hotspots of municipalities were then identified by the Local Moran's Index (S1 Fig). The first hotspot ($c_1$) is located in the central-western region of Mexico. It includes 49 municipalities in Jalisco, Michoacán, Guanajuato, and Querétaro. The second hotspot ($c_2$) is in the central-eastern region of Mexico comprising 12 municipalities in Puebla and Veracruz. The third hotspot ($c_3$) is in the northern region and includes six municipalities in Sonora. Finally, the fourth hotspot ($c_4$) is in the southeastern region and consists of 17

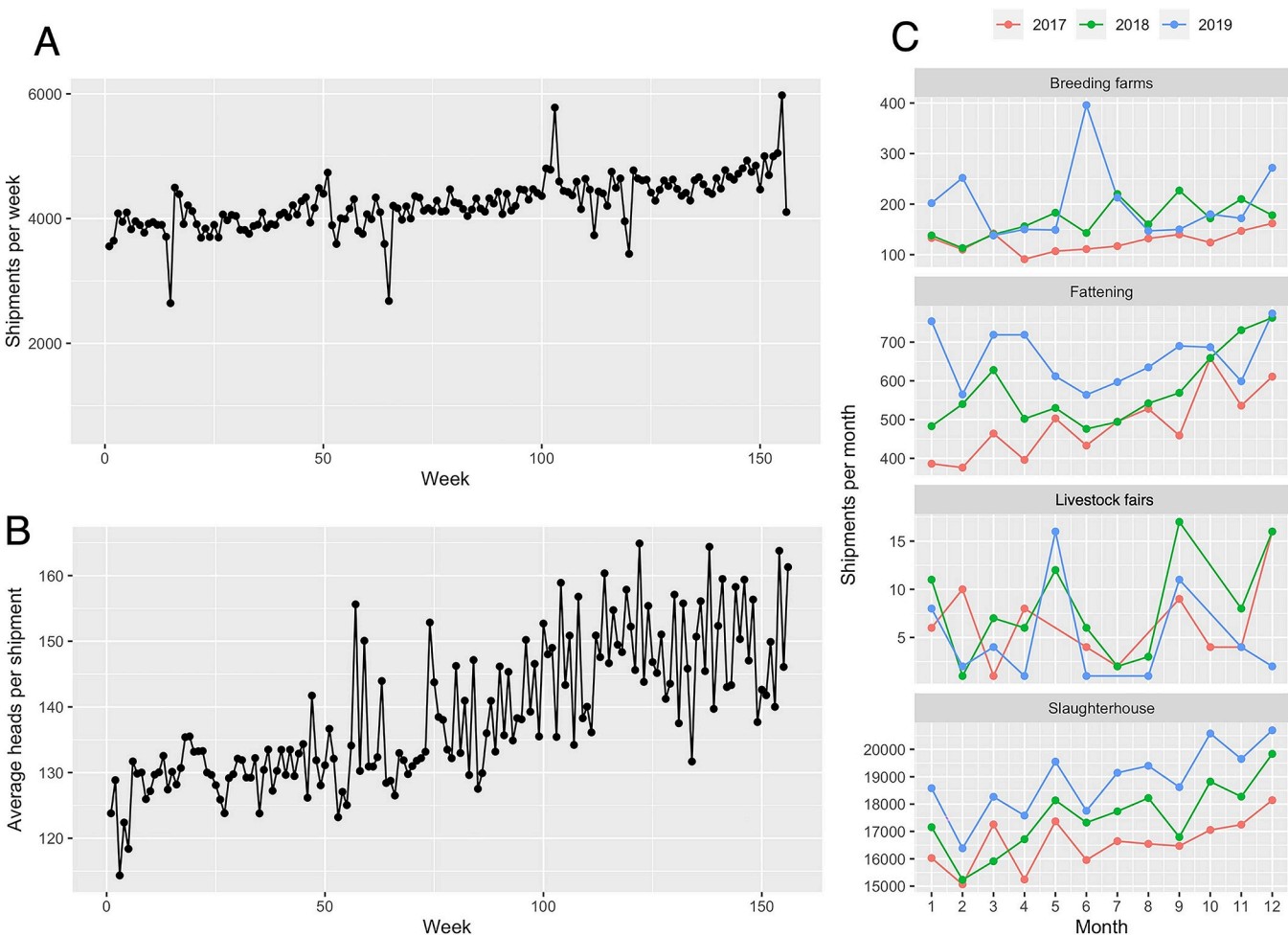

**Fig 1. Temporal trend of swine movements in Mexico from 2017 to 2019.** Weekly frequency of pig shipments (A), average size of shipments (head per shipment) (B) and the seasonality of the monthly frequency of the shipments divided by the purpose of movement (C).

municipalities in Yucatan. The rest of the municipalities of Mexico had no hotspots with high swine density. The names of the municipalities within each hotspot and their respective states are listed in S1 Table. These four hotspots are estimated to concentrate approximately 55.7% of the pigs in Mexico. In addition, these hotspots revealed a high flow of incoming and outgoing movements (Table 2).

As shown in Fig 3A, different movement flow patterns of the pigs were identified between all hotspots. In both $c_1$ and $c_2$, the outgoing pig shipments individually exceeded to their incoming shipments. The outgoing movements from c1 represent 61.4% of all its movements, while for c2, they account for 86.3%. On the other hand, $c_3$ and $c_4$ were characterized by their internal movements, suggesting significant internal trade of pigs within the municipalities comprising each hotspot. These internal movements accounted for 90.1% and 69.5% of the total pig movement in these hotspots. Finally, the movement of pigs between these hotspots was not a significant factor in terms of quantity.

The flows patterns for different purposes, such as shipments to slaughterhouses or for fattening, were similar in terms of origin and destination, but there were differences in the numbers of shipments for each hotspot (Fig 3B–3E).

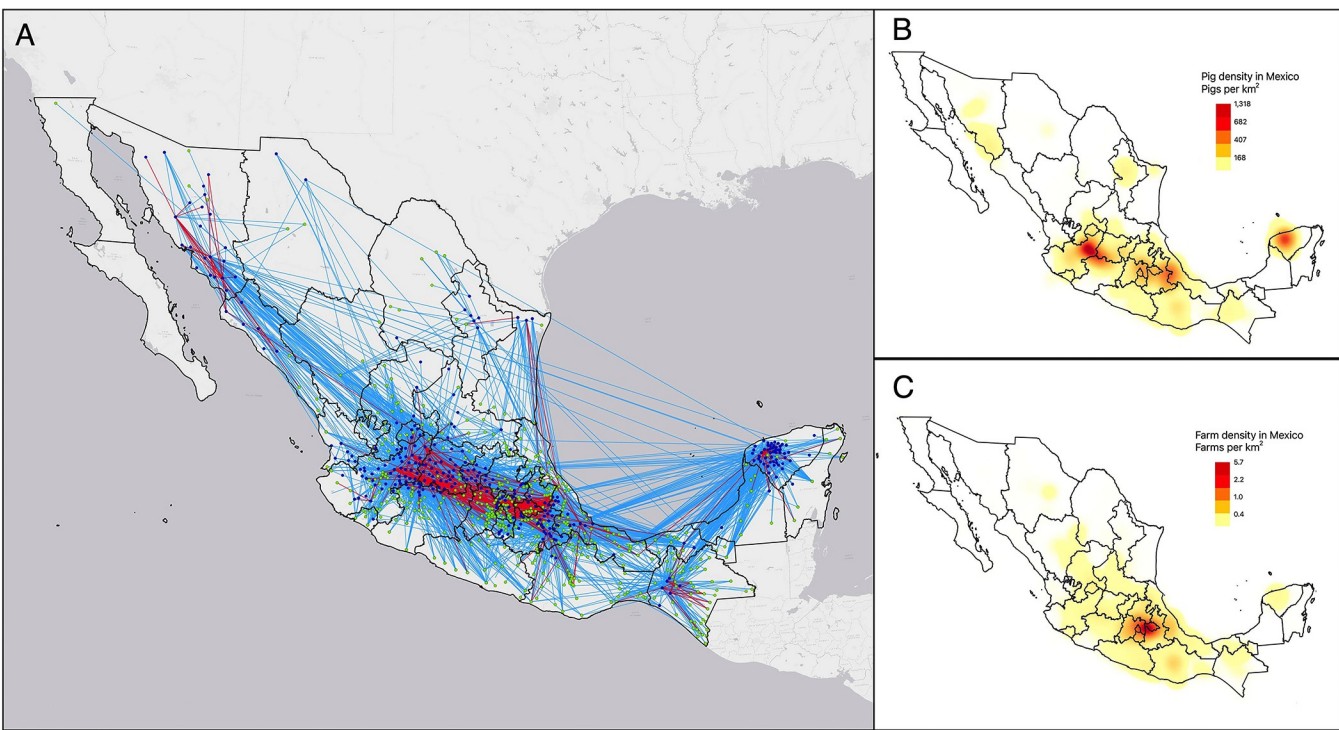

**Fig 2. The distribution of swine movements and pig density in Mexico between 2017 and 2019.** The nodes represent the centroids of the municipalities with outgoing shipments (navy blue) and incoming shipments (green) (A). The red lines represent edges with a high frequency of pig shipments (> 310 shipments during the study period [95th percentile]), while the blue lines represent regular movements (range: 3–310). Edges and nodes with fewer than 3 shipments are not displayed to improve visual clarity. Kernel density maps of swine population (B) and farm distribution in 2019 (C). Note: The map of the administrative boundaries was obtained from the Marco Geoestadistico, provided by the Instituto Nacional de Estadistica y Geografia (INEGI). Source: https://www.inegi.org.mx/temas/mg/#descargas. To prepare the base map, we used the QuickMapServices plugin in QGIS, obtaining the data from NextGIS. The source link can be accessed at: https://qms.nextgis.com/.

### Description of the swine movement network

The complete network of swine movements connected 54% (1,334) of all municipalities. At the end of 2019, there were 93,174 pig farms registered in PGN, with 25.3% of these farms are located in municipalities without a connection to the network. Therefore, there is a high probability that these farms did not have a report of pig movements in the SNAM.

The complete network of swine movements had a total of 662,255 shipments (edges) grouped into 8,328 pairs of movements between nodes (Table 3). The sub-network of swine movements to slaughterhouses connected 44.6% of the municipalities and was the most extensive network in terms of geographical coverage. The sub-network for breeding ranked second (33.9% of the municipalities), followed by fattening (23.5%) and finally livestock fairs (3.8%).

A total of 92 municipalities connected to the complete network demonstrated the presence of feedback loops or internal movements. This represents 6.9% of all municipalities. These feedback loops account for 11.6% of all pig shipments and 6.8% of all pigs shipped during the study period. Therefore, we included the looped shipments in subsequent analyses to preserve representativeness.

On average, 79.5 shipments occurred between each pair of nodes in the complete network of swine movements. The average of pig shipments between nodes exceeded the median, indicating a highly right-skewed distribution in all networks of swine movements (Table 3). The

**Table 2. Description of swine shipments and pig population in hotspots of municipalities with high swine density and the rest of the municipalities in Mexico.**

| Variables | Hotspots[a] | | | | | Other municipalities |
|---|---|---|---|---|---|---|
| | $c_1$ | $c_2$ | $c_3$ | $c_4$ | Total | |
| Number of municipalities | 49 | 12 | 6 | 17 | 84 | 2,386 |
| *Swine population[b]* | | | | | | |
| Swine farms | 3,086 | 322 | 433 | 237 | 4,078 | 89,096 |
| Swine farm density (farms/km²) | 0.12 | 0.1 | 0.01 | 0.06 | 0.07 | 0.05 |
| Swine population | 4,270,602 | 923,750 | 1,817,753 | 455,255 | 7,467,360 | 5,939,945 |
| Swine poulation density (km2) | 170.9 | 279.2 | 65.2 | 116.0 | 124.3 | 3.11 |
| Mean Local Moran Index | 7.73 | 4.56 | 1.21 | 2.24 | 5.7 | 0.03 |
| *Shipments mobilized* | | | | | | |
| Internal | 64,167 | 10,290 | 47,192 | 17,051 | 138,700 | 146,540 |
| Incoming | 40,722 | 3,341 | 21,490 | 22,628 | 88,181 | 311,648 |
| Outgoing | 102,103 | 64,988 | 5,165 | 7,480 | 179,736 | 197,279 |
| *Shipped pigs* | | | | | | |
| Internal | 7,608,005 | 350,285 | 7,330,123 | 2,509,488 | 17,797,901 | 15,984,463 |
| Incoming | 7,722,874 | 412,388 | 6,483,160 | 3,194,527 | 17,812,949 | 49,179,333 |
| Outgoing | 12,055,324 | 9,369,004 | 2,959,029 | 678,621 | 25,061,978 | 33,379,875 |

[a] S1 Table lists the names of the municipalities that make up the hotspots.

[b] Swine population data updated to 2019.

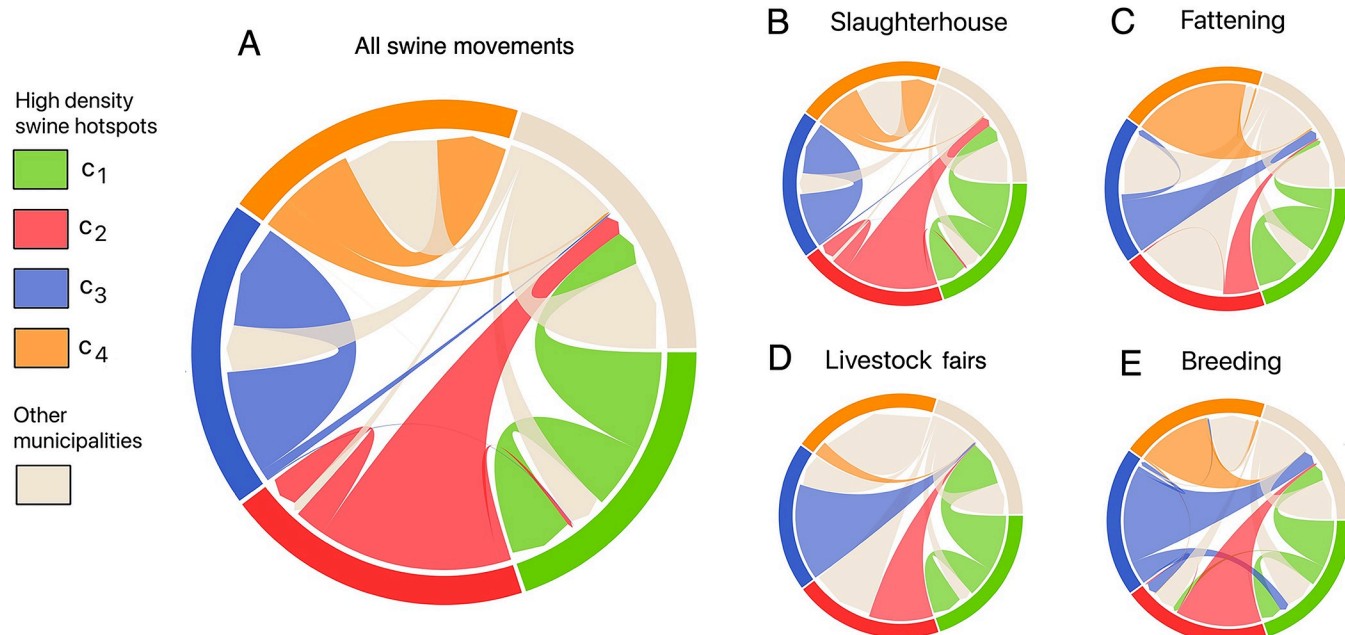

**Fig 3. Swine movement flow between hotspots in Mexico.** Flow of all swine movements (A) and separated by purposes: slaughterhouse (B), fattening (C), livestock fairs (D) and breeding (E). The thickness of the arcs corresponds to the proportion of shipments compared to the total, both origin and destination. The color indicates their hotspot of origin.

**Table 3. Summary of attributes and metrics of swine movement networks.**

| Variables | Complete network | Sub-networks | | | |
|---|---|---|---|---|---|
| | | Slaughterhouse | Fattening | Livestock fairs | Breeding farms |
| *Network atributes* | | | | | |
| Number of municipalities (nodes) | 1,334 | 1,101 | 581 | 93 | 838 |
| Nodes with loops | 92 | 78 | 38 | 4 | 29 |
| Number of shipments (edges) | 662,255 | 635,397 | 20,678 | 203 | 5,977 |
| Pairs of movements between nodes | 8,328 | 6,329 | 1,215 | 103 | 2,014 |
| *Shipments* | | | | | |
| Median | 3 | 6 | 2 | 1 | 1 |
| Mean | 79.5 | 100.4 | 17 | 2 | 3 |
| 95th percentile | 310 | 412.6 | 67.6 | 5 | 7 |
| Maximum | 12,164 | 12,125 | 2,029 | 13 | 254 |
| *Euclidean distance (edge length), km* | | | | | |
| Median | 199.3 | 188.5 | 155.4 | 111.5 | 231.5 |
| Mean | 300 | 262.7 | 200.1 | 205.8 | 405.9 |
| 95th percentile | 948.2 | 731 | 966 | 859.5 | 1,349.70 |
| Maximum | 2,375.50 | 2,081.30 | 1,381.80 | 1,339.30 | 2,375.50 |
| *Graph level metrics* | | | | | |
| Nerwork density (%) | 0.14% | 0.10% | 0.02% | 0.00% | 0.03% |
| Reciprocity (%) | 6.46% | 6.20% | 9.20% | 9.70% | 3.40% |
| Transitivity | 0.13 | 0.14 | 0.08 | 0.03 | 0.03 |
| Diameter | 10 | 10 | 11 | 4 | 11 |
| Assortativity | 0.02 | 0.03 | 0.03 | 0.09 | -0.02 |
| *Node-level metrics (In-degree)* | | | | | |
| Mean | 3.34 | 2.54 | 0.48 | 0.4 | 0.81 |
| 95th percentile | 16 | 13 | 3 | 0 | 4 |
| Maximum | 101 | 96 | 25 | 10 | 17 |
| *Node-level metrics (Out-degree)* | | | | | |
| Mean | 3.34 | 2.54 | 0.48 | 0.04 | 0.81 |
| 95th percentile | 19 | 14 | 2 | 0 | 1 |
| Maximum | 301 | 219 | 108 | 25 | 191 |
| *Node-level metrics (Betweenness)* | | | | | |
| Mean | 398.7 | 295 | 41.56 | 0.17 | 56.25 |
| 95th percentile | 743 | 326 | 0 | 0 | 0 |
| Maximum | 49,939 | 50,839 | 13,353 | 264 | 16,047 |

pig shipments were transported an average distance of 300 km. Except for pig shipments to livestock fairs, statistically significant differences were observed in the average distance between nodes according to the purpose of movement (F = 173.8, P < 0.001).

## Description of network level metrics

Table 3 summarizes the metrics results for all networks of swine movements. The study found that the density of swine networks ranged from 0.001% to 0.14%, with the most significant proportion of bidirectional connections in sub-networks for fattening and livestock fairs. The sub-network to slaughterhouses showed the highest transitivity value, suggesting interconnected municipalities. There was no evidence of preferential connections between municipalities with similar pig populations.

In the temporal analysis, the complete network and slaughterhouse sub-network gradually increased in density but showed no significant changes in reciprocity, transitivity, or diameter values, as illustrated in S2 Fig. In fattening and breeding sub-networks, intermittent peaks in reciprocity and transitivity values suggest that short periods of swine movement generate more symmetrical and cohesive temporal interactions between municipalities.

### Description of node-level metrics

In the complete network of swine movements, the municipalities with the highest in-degree were located in the central region of Mexico, mainly in municipalities of $c_1$. Furthermore, in all hotspots ($c_1$, $c_2$, $c_3$, $c_4$), there were municipalities with the highest out-degree, which also had a high betweenness. The distribution of the degree centrality and betweenness metrics results in the complete network of swine movements are show in more detail in Fig 4.

A general trend was observed in all networks, with municipalities showing high out-degree values and betweenness values. A similar trend was observed between in-degree and betweenness; however, only a few municipalities presented high values for both metrics. Although some changes in the value of the metrics were observed in all sub-networks, the same trend in the results and their location in the hotspots was maintained.

For all networks, there were no consistent patterns; municipalities with high in-degree were not also systematically identified as hubs, nor were municipalities with high out-degree generally identified as authorities, suggesting greater complexity in the structure of the movement networks.

### Correlation between node metrics and indicators of swine population and commercial facilities

Moderately strong positive correlations ($\rho \geq 0.4$ and $< 0.6$, $p < 0.001$) were detected between population indicators and node metrics in all networks of swine movements, except in the sub-network of transport to livestock fairs. Swine density and number of technified farms showed the strongest correlations. In contrast, we observed weak correlations with commercial facilities indicators. S2 Table shows the complete results of the correlation analysis.

### Detection of commercial communities

We applied the random walk selection of the *Walktrap* algorithm, and we identified 11 communities, of which six were large (Fig 5). The weighted selection of the algorithm identified the same number of large communities but reduced the number of nodes per community by eliminating weaker edges or those with fewer shipments (S3 Table).

### Discussion

This study provides the first comprehensive overview of swine movement patterns and trade networks in Mexico, encompassing movements to farms, slaughterhouses, and livestock fairs. Previous research has focused on a small fraction of pig movement to slaughterhouse [66]. The use of official records of swine movements enables the generation of data that can inform decision-makers regarding the monitoring of disease control and biosecurity measures [67].

The most common reason for the movement of pigs in Mexico is for slaughter, a finding that is not consistent with previous research in other countries [68–70]. The high number of shipments to the slaughterhouse can be attributed to the volume of production of technified farms and their biosecurity protocols, which restrict the movements of pigs between farms or facilities [17, 71, 72]. This discrepancy indicates the presence of a unique pattern of movement within the Mexican pig sector, warranting further investigation.

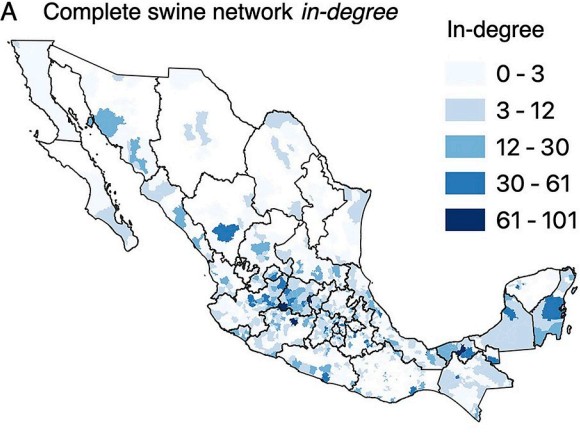

A  Complete swine network *in-degree*

In-degree
- 0 – 3
- 3 – 12
- 12 – 30
- 30 – 61
- 61 – 101

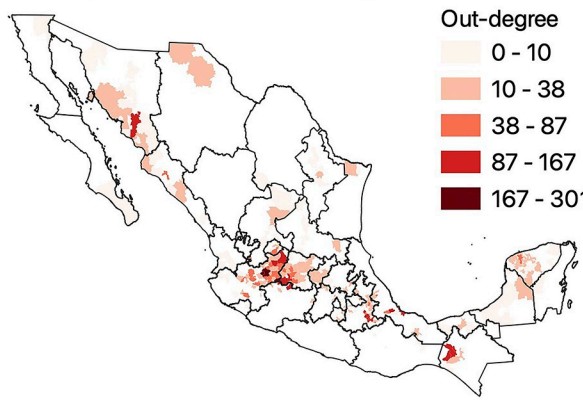

B  Complete swine network *out-degree*

Out-degree
- 0 – 10
- 10 – 38
- 38 – 87
- 87 – 167
- 167 – 301

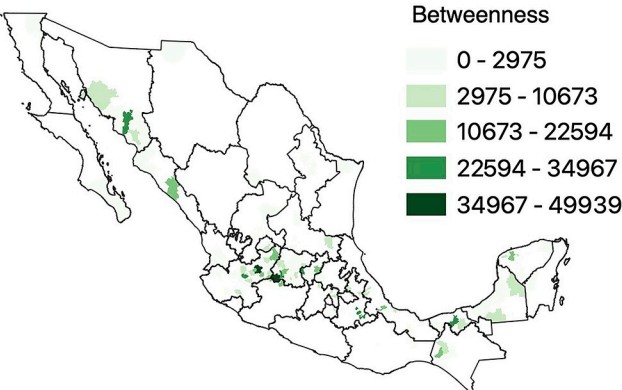

C  Complete swine network *betweenness*

Betweenness
- 0 – 2975
- 2975 – 10673
- 10673 – 22594
- 22594 – 34967
- 34967 – 49939

**Fig 4. Distribution of node-level metrics for the complete network of swine movements.** *In-degree* (A), *out-degree* (b), and *betweenness* (c) values. Note: The map of the administrative boundaries was obtained from the Marco Geoestadistico, provided by the Instituto Nacional de Estadistica y Geografia (INEGI). Source: https://www.inegi.org. mx/temas/mg/#descargas.

Our results confirm that swine movements in Mexico increased in terms of both the frequency and the size of shipments over the study period. This trend can be attributed to the growing domestic pork demand in Mexico and the expanding volume of exports [73–75]. It is

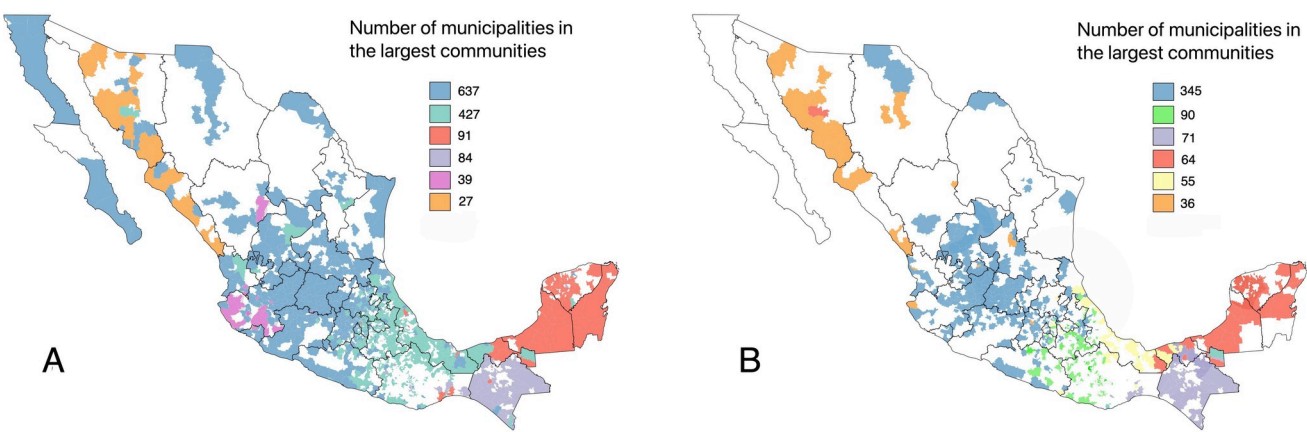

**Fig 5. Members of the large communities identified by the *Walktrap* algorithm.** The selection process of random walks (A) and weighted by the number of shipments (B). To improve the readability of the map, small communities are not displayed. Note: The map of the administrative boundaries was obtained from the Marco Geoestadistico, provided by the Instituto Nacional de Estadistica y Geografia (INEGI). Source: https://www.inegi.org.mx/temas/mg/#descargas.

expected to continue and possibly accelerate in the future. As a consequence, new challenges in terms of biosecurity, logistics and disease control throughout the supply chain will probably emerge [76–80].

While there has been an improvement in SNAM coverage with more farmers registering their pig movements, there is still a significant gap due to a lack of awareness, resources, and renounce to adopt digital platforms. These challenges need to be addressed to encourage greater participation in traceability in the Mexican pork industry.

The seasonal movement patterns of pigs found in our study are consistent with the seasonality of swine production in Mexico [81]. The seasonal movement patterns are characterized by a high season towards the end of the year (October-December) due to the festivals and celebrations that occur during this period. The period of lowest pig movements occurs in spring (March-April), which coincides with a religious period when the Catholic population in Mexico reduces its meat consumption [82]. Another possible explanation for these seasonal movement patterns is the adaptive production strategy employed by the pig farms, as well as market demand fluctuations, climatic conditions, and feed availability [83, 84]. This could be interesting to explore in future research.

Our analysis of the geographic distribution of pig farms identified four hotspots of high-density municipalities ($c_1$, $c_2$, $c_3$ and $c_4$), considered the main centers of swine production in Mexico. These hotspots represent critical points for pig movement, especially for fattening, breeding, or livestock fairs. Multiple factors, including geography, demand for pork, and production practices, influence these profiles. For example, pig production in the hotspots of Sonora and Yucatán ($c_3$ and $c_4$) focuses on fattening and slaughtering pigs within the same region, with a significant portion of the pork meat being exported, particularly to Asia [75, 85]. In contrast, the hotspots identified in the central region of Mexico ($c_1$ and $c_2$) focus on fattening and transportation of live pigs to other municipalities, mainly to the most populated areas of Mexico.

Analysis of the complete network of swine movements revealed a complex interconnection between municipalities in Mexico. This is the result of the exchange of pigs within the hotspots ($c_1$, $c_2$, $c_3$, and $c_4$) and to the rest of the municipalities. The low density in all networks of swine movements could indicate the presence of limited or underutilized edges between origin and destination municipalities.

The network diameter and the average Euclidean distance between nodes indicate a comprehensive geographic coverage of swine movements. The geographic coverage of this network, which is predominantly local, particularly associated to the hotspots, could explain the low density of connections, thereby facilitating more efficient mobilization coverage. However, we also identified outliers in the Euclidean distance data. These outliers could be due to exceptional movements, such transport for high-value pigs.

This interconnection poses health and biosecurity challenges due to the high concentration of farms and shipments, mainly in the swine production centers in Mexico [42, 86–88]. Although a low network density could limit the spread of disease, the existing links between nodes due to the regular movement of pig shipments represents a potential pathway for the spread of disease [89].

The most frequent movements within the complete network are directed towards slaughterhouse, traditionally considered "dead-ends" regarding disease transmission. Despite this, there is evidence that pathogens are being spread to farms near slaughterhouses or through transport vehicles, especially those transmitted by indirect contact or mechanical vectors [90, 91]. This observation underscores the necessity of enhancing surveillance and biosecurity measures at slaughterhouses, especially in municipalities associated with the swine density hotspots.

In sub-networks of swine movements, bidirectional connections were detected during short periods in the fattening and breeding movements. These recurrent exchanges suggest that these pig movements occur only when there is a specific association between breeding and finishing locations or involve multi-site pig farms [39].

Our analysis of swine movement network reveals several limitations that must be considered for an accurate interpretation of the results. One such limitation is the inability to track movements at the local level. This is evidenced by the low number of loops found in the municipalities, indicating that internal mobilization was not reported to SNAM. The absence of such local data could result in an underestimation of the complexity and real extent of swine mobilization networks.

Another limitation is the lack of information to identify the type of pig farms involved in the mobilization network. This includes whether the farms are technified, semi-technified or backyard. Although the correlation analysis indicated that technified farms may have influenced the configuration of outgoing and incoming connections of the municipalities, further analyses are required to determine the impact of other farms in the network.

The observation of outliners in shipments larger than 500 head to the SNAM may introduce bias in the identification of critical network nodes and affect network metrics. Although previous studies have documented multiple pig movements in a single transaction [42, 92], it is important to improve the quality of data collection for these movements, even if this practice is not typical.

## Conclusions

The study examines swine movement patterns in Mexico, using official movement records as the data source. Provides a comprehensive overview of the sector, capturing the general trends and dynamics of these movements. The analysis is particularly relevant in Mexico, where the swine sector faces threats from endemic and transboundary diseases such as ASF. While the results are limited to a specific period and primarily reflect the large-scale swine industry, which is less likely to reflect smallholder movements, further research is needed to better understand their dynamics.

The results of this study could help enhance the accuracy of official movement records and improve the assessment of national disease transmission risks. Furthermore, the study

identifies significant nodes and commercial communities in Mexico that can be targeted in disease response planning. Our findings indicate that the Mexican swine sector is characterized by the concentration of four high-density swine production centers, which have a significant influence on the swine movement network and are critical for establishing strategic surveillance and disease control points.

## Supporting information

**S1 Fig. Hot and cold spot map illustrating the spatial distribution of municipalities with high swine density in Mexico.** This map highlights areas in red indicating concentrated swine populations (hot spots) and regions in navy blue with lower swine density (cold spots). The map reveals areas in transition, with Low-High (light blue) and High-Low (light red) gradients. The outlined squares in the figure denote the location of swine production centers ($c_1$, $c_2$, $c_3$, and $c_4$). Note: The map of the administrative boundaries was obtained from the Marco Geoestadistico, provided by the Instituto Nacional de Estadistica y Geografia (INEGI). Source: https://www.inegi.org.mx/temas/mg/#descargas.
(TIF)

**S2 Fig. Metrics of the complete network of swine movements and sub-networks by purpose of movement calculated weekly during 2017–2019.** The following metrics are described: density (A), reciprocity (B), transitivity (C), and diameter (D).
(TIF)

**S1 Table. List of Municipalities comprising the four hotspots with high swine density identified in Mexico.**
(PDF)

**S2 Table.** *Spearman* **range correlation analysis between the metrics of nodes and indicators of swine population and commercial facilities.**
(PDF)

**S3 Table. Results of detection of commercial communities in the complete network of swine movements from 2017 to 2019.**
(PDF)

## Acknowledgments

We acknowledge the support of the Servicio Nacional de Sanidad, Inocuidad y Calidad Agroalimentaria (SENASICA) for providing the swine movement data essential for our analysis.

## Author Contributions

**Conceptualization:** Alejandro Zaldivar-Gomez, Jose Pablo Gomez-Vazquez, Beatriz Martínez-López, Gerardo Suzán, Oscar Rico-Chávez.

**Formal analysis:** Alejandro Zaldivar-Gomez, Jose Pablo Gomez-Vazquez.

**Investigation:** Alejandro Zaldivar-Gomez.

**Methodology:** Alejandro Zaldivar-Gomez, Jose Pablo Gomez-Vazquez.

**Project administration:** Oscar Rico-Chávez.

**Supervision:** Beatriz Martínez-López, Oscar Rico-Chávez.

**Validation:** Beatriz Martínez-López.

**Visualization:** Alejandro Zaldivar-Gomez.

**Writing – original draft:** Alejandro Zaldivar-Gomez.

**Writing – review & editing:** Jose Pablo Gomez-Vazquez, Beatriz Martínez-López, Gerardo Suzán, Oscar Rico-Chávez.

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
