## [Decision Letter · Decision Letter 0]

4 Apr 2024

PONE-D-23-44053Analysis of the swine movement network in Mexico: A perspective for disease prevention and control.PLOS ONE

Dear Dr. Rico-Chávez,

Thank you for submitting your manuscript to PLOS ONE. After careful consideration, we feel that it has merit but does not fully meet PLOS ONE’s publication criteria as it currently stands. Therefore, we invite you to submit a revised version of the manuscript that addresses the points raised during the review process.

We look forward to receiving your revised manuscript.

Kind regards,

Jean-François Carod

Academic Editor

PLOS ONE

Journal Requirements:

3. a. For studies reporting research involving human participants, PLOS ONE requires authors to confirm that this specific study was reviewed and approved by an institutional review board (ethics committee) before the study began. Please provide the specific name of the ethics committee/IRB that approved your study, or explain why you did not seek approval in this case.

b. Please provide additional details regarding participant consent. In the ethics statement in the Methods and online submission information, please ensure that you have specified what type you obtained (for instance, written or verbal, and if verbal, how it was documented and witnessed). If your study included minors, state whether you obtained consent from parents or guardians. If the need for consent was waived by the ethics committee, please include this information.

Alejandro Zaldivar-Gomez is supported by the Programa de Becas para Estudios de Posgrado, approved by the Consejo Nacional de Humanidades, Ciencias y Tecnologías (Conahcyt) (Grant number: 750549).

5. Thank you for uploading your study's underlying data set. Unfortunately, the repository you have noted in your Data Availability statement does not qualify as an acceptable data repository according to PLOS's standards.

7. PLOS requires an ORCID iD for the corresponding author in Editorial Manager on papers submitted after December 6th, 2016. Please ensure that you have an ORCID iD and that it is validated in Editorial Manager. To do this, go to ‘Update my Information’ (in the upper left-hand corner of the main menu), and click on the Fetch/Validate link next to the ORCID field. This will take you to the ORCID site and allow you to create a new iD or authenticate a pre-existing iD in Editorial Manager. Please see the following video for instructions on linking an ORCID iD to your Editorial Manager account: https://www.youtube.com/watch?v=_xcclfuvtxQ

Additional Editor Comments:

Thanks for your submission "Analysis of swine movement network in Mexico: A perspective for disease prevention and control" , please consider both of the reviewers comments and take action for a resubmission including the major changes requested.

Yours sincerelly,

Dr Jf Carod

Reviewers' comments:

Reviewer's Responses to Questions

**Comments to the Author**

1. Is the manuscript technically sound, and do the data support the conclusions?

Reviewer #1: Yes

Reviewer #2: Partly

2. Has the statistical analysis been performed appropriately and rigorously? 

Reviewer #1: Yes

Reviewer #2: Yes

3. Have the authors made all data underlying the findings in their manuscript fully available?

Reviewer #1: Yes

Reviewer #2: Yes

4. Is the manuscript presented in an intelligible fashion and written in standard English?

Reviewer #1: Yes

Reviewer #2: Yes

5. Review Comments to the Author

**Reviewer #1:** The article provides relevant information about the swine network in Mexico, which is important for national improvements of the sanitary system and general risk analysis. The article provides extensive network analysis. My general recommendation is to condense the scientific text to enhance readability and maintain focus.

Dear authors, please find below some recommendations based on the text provided, aimed at enhancing the quality of your hard work:

Line 57: Could you explain better this percentages? The pig population was counted based on the number of farms or number of pigs (could you include the numbers of the estimation), percentage of production refers also to metric tons of meat or number of slaughtered animals?

Line 110: Municipal is a third or fourth level political organization. Federal, State and Municipal? Could you explain this.

Line 143: Spanish original names could be italic, or a short English name followed by the Spanish on parenthesis.

Line 216: Please consider indicate that you used hubs and authorities score, just for general audience readers.

Line 223: What about the semi-technified and backyard?

Line 228-241: Could you merge these two paragraphs.

Line: 238: Could you rephrase for better readability, why 20?

Line 265: Could you include the percentage of reduction.

Line 264: According to Fig.1C, the movement from livestock fairs and fattening reduces between Jun, Jul, Ago any explanation for this reduction (On the discussion the reduction is explained around Mar-Apr).

Line 278: According to Fig. 1C the number of shipments per month to livestock fairs, are extremely low, considering the 600.000 movements, it shows that the clients of that fairs should be backyard producers without registering those movements, is that appreciation, correct? Could you comment about that?

Line 288 Fig. 2. Could you comment about the long-distance movements (e.g. Line Merida – Chihuahua or Hemosillo >2500 km).

Table 2: Please consider including on Swine populationb the density (pigs/farms per Km2) to have a direct density comparison between hotspots, also could you consider including the local Moran index on the text or on the table for the hotspots, it could be useful (I suppose that local index should be higher than global index).

Line 323: When superimposing Fig2 and SM1 seems that there are some movements close to the c3 and c4 hotspots, please consider looking further if they are related or just errors on the data as my previous comment about line 288.

Line 317-323: Please consider include a percentage to the 15,148 considering the reference of total movement in the hotspots, some are explained as numbers and other as percentages, that makes harder to read.

Line 333: Please consider explain what you refer when talking about similarities and discrepancies.

Line 338: Please consider indicate the total number of farms registered on the PGN. Do you consider that they are backyard farms?

Table 3: Euclidean distance: 95th percentile and maximum distances are quite high biological plausibility of terrestrial transport, consider checking the validity of those movements, or maybe they are aerial transport maybe for high valuable pigs.

Line 439: Please consider rephrasing the loops explanation.

Line 366: Please consider which was the first assumption and based on what analysis.

404: Please consider separating methods and results in this paragraph.

Line 420: Please consider rephasing this paragraph.

Line 425: Please consider explaining better why on the map there are NA. My understanding is that 11 communities were identified, the largest 6 are plotted on the map and the other 5 (weaker) are plotted as NA.

Line 428: Please consider providing a clearer explanation of “geographically clearly defined” when looking at the map (Fig. 5B) as the colours appear quite scattered, particularly the blue ones. Additionally, regarding the “correspondence” of hotspots to the communities: C4 corresponds to the “red” community, but one municipality is on the opposite northeast side of the country. C2 is divided between the “blue”, “green”, “yellow”, NA (hard to tell); C1 is within the blue community, with some NA as well. Maybe a count of the match could help or use “approximate” in reference to the hotspots.

Line 459: Please consider that this paragraph only addresses the industrial perspective, whereas backyard producers might find it easier to slaughter animals on-site. There is evidence of movements to and from markets and maintaining them for ~1o monthly movements seems economically inefficient (please consider include the number of those markets). It is possible that other users of those livestock fairs are not registering their movements, as indicated in 578. Perhaps it could be interesting to keep this as an open question in the discussion for further research projects.

Line 467: Please consider that movements to slaughterhouses are usually considered as “dead-ends” with respect to disease transmission. How the network without dead ends would look like for using in a disease transmission model?

Line 476: Please consider rephrasing this paragraph.

Line 551: Please consider resume this long paragraph.

**Reviewer #2: **The paper "Analysis of swine movement network in Mexico: A perspective for disease prevention and control" uses social network analysis (SNA) methods to investigate patterns in Mexico's officially recorded swine movement data. The authors provide the analysis results through plots and tables and discuss their implications. While this kind of analysis is not new, the authors claim this is an original application of SNA in Mexico's swine movement networks. Here is my feedback based on what I understood from reading the paper.

1. The analysis is presented at the municipality level (as nodes) and not the farm level. Hence, there are limitations in interpreting the results. These limitations should be clarified to the readers.

2. The author's estimate of about 95.9% being attributed to slaughterhouse movements could be misleading if it is just based on shipment counts rather than pig population. Heads in fattening shipments are almost four times more than slaughterhouse shipments. This should be clarified in the wording.

3. Some of the statements in the paper are too trivial. For example, statements such as "movements occur in the same areas where swine production is concentrated..." do not provide any interesting insight. The authors should limit these statements and make the write-up concise.

4. Lines 191-193: The authors use and report Euclidean distance to measure distances between municipalities/nodes. However, shipment distances computed from road networks will likely better fit this context.

5. The discussion section is too long, where the authors provide a lot of commentary, but for the reader, it is difficult to extract information that could be used in later studies. For example, what are the top 2 or 3 insights/findings that are novel contributions of this paper?

6. Lines 262-264: Grammar issues.

6. PLOS authors have the option to publish the peer review history of their article (what does this mean?). If published, this will include your full peer review and any attached files.

Reviewer #1: **Yes: **Alfredo Acosta

Reviewer #2: No

---

## [Author Response · Author response to Decision Letter 0]

10 Jun 2024

5. Review Comments to the Author

Reviewer #1: The article provides relevant information about the swine network in Mexico, which is important for national improvements of the sanitary system and general risk analysis. The article provides extensive network analysis. My general recommendation is to condense the scientific text to enhance readability and maintain focus.

Dear authors, please find below some recommendations based on the text provided, aimed at enhancing the quality of your hard work:

Line 57: Could you explain better this percentages? The pig population was counted based on the number of farms or number of pigs (could you include the numbers of the estimation), percentage of production refers also to metric tons of meat or number of slaughtered animals?

Thank you for your feedback. We have included the total pig population and pork production in Mexico. Please see the lines 56 – 59.

Line 110: Municipal is a third or fourth level political organization. Federal, State and Municipal? Could you explain this.

We made the changes suggested. Please see the lines 110-112.

Line 143: Spanish original names could be italic, or a short English name followed by the Spanish on parenthesis.

We used short English names followed by the Spanish names in parentheses. All changes were made in lines 24, 81, 85, 129, and 145.

Line 216: Please consider indicate that you used hubs and authorities score, just for general audience readers.

We have added an explanation about this topic and rephrased the text. Please see the lines 217 – 219.

Line 223: What about the semi-technified and backyard?

The swine population variables include data from technified, semi-technified, and backyard farms. However, we created a subset of technified farm because these types of farms frequently report to the SNAM database. Distinction between these categories of farms is explained in lines 224 - 227.

Line 228-241: Could you merge these two paragraphs.

We have rewritten the paragraph to improve readability. See lines 231-238.

Line: 238: Could you rephrase for better readability, why 20?

We observed that some communities ranged from 1 to 3 nodes, while the others had more than 27 nodes. To address this discrepancy, we established a cut-off point of 20 nodes, categorizing communities with fewer than 20 nodes as small and those with 20 or more nodes as large. Please see the explanation in the manuscript in lines 236-238.

Line 265: Could you include the percentage of reduction.

We made the adjustment in the manuscript. Please see the line 261-263.

Line 264: According to Fig.1C, the movement from livestock fairs and fattening reduces between Jun, Jul, Ago any explanation for this reduction (On the discussion the reduction is explained around Mar-Apr).

In the section of discussion, we have focused on explaining the reduction in swine movements observed during the March to April period. We attribute this decline to the influence of religious traditions, as corroborated by the available data.

The observed pattern of movements to fairs from June to August may be influenced by other factors, such as the scheduling of fairs or an economic factor, but we don't have the data to discuss this pattern. We made a clarification in the manuscript. Please see the lines 442 - 447

Line 278: According to Fig. 1C the number of shipments per month to livestock fairs, are extremely low, considering the 600.000 movements, it shows that the clients of that fairs should be backyard producers without registering those movements, is that appreciation, correct? Could you comment about that?

Although it is very likely that backyard producers do not participate in the registration of swine movements, we are currently unable to identify the type of farm in movement data. We compared the number of farms from the National Livestock Register (PGN) with network metrics and found a correlation specifically with technified farms. This provides indirect evidence of the influence that technified farms have on the configuration of the network. However, further research is needed in the future.

In the discussion we noticed these limitations. See the lines 495 – 499.

Line 288 Fig. 2. Could you comment about the long-distance movements (e.g. Line Merida – Chihuahua or Hemosillo >2500 km).

Long-distance movements are exceptional and involve a small number of animals and shipments. They are associated with the sale of animals for breeding purposes or high-value pigs. We have included an explanation of long-distance movements in the discussion to ensure the reader is fully aware of these factors. See the lines 468-470.

Table 2: Please consider including on Swine population the density (pigs/farms per Km2) to have a direct density comparison between hotspots, also could you consider including the local Moran index on the text or on the table for the hotspots, it could be useful (I suppose that local index should be higher than global index).

Thank you for the suggestion. We included the density of farms and pigs per km2 in Table 2. Regarding the local Moran Index, we calculate the average value per hotspot, as it is calculated individually for each municipality. The local Moran index is indeed higher than the value of the global index (I = 0.229, P < 0.05), indicating spatial clustering in the dataset.

Line 323: When superimposing Fig2 and SM1 seems that there are some movements close to the c3 and c4 hotspots, please consider looking further if they are related or just errors on the data as my previous comment about line 288.

We have reviewed the database, and we can confirm that the movements observed near the c3 and c4 hotspots are correctly recorded. As we mentioned in the previous commentary, these movements are exceptional with few animals and are due to shipments of high value pigs. We have included an explanation of this long-distance movement in the discussion. See the lines 468-470.

Line 317-323: Please consider include a percentage to the 15,148 considering the reference of total movement in the hotspots, some are explained as numbers and other as percentages, that makes harder to read.

We have made the suggested adjustments. See lines 345 – 349.

Line 333: Please consider explain what you refer when talking about similarities and discrepancies.

The differences and similarities were clarified in the manuscript and the paragraph was reduced for better reading on lines 327 -329.

Line 338: Please consider indicate the total number of farms registered on the PGN. Do you consider that they are backyard farms?

We have made the suggested adjustments. Please see the line 333.

Table 3: Euclidean distance: 95th percentile and maximum distances are quite high biological plausibility of terrestrial transport, consider checking the validity of those movements, or maybe they are aerial transport maybe for high valuable pigs.

We checked the database and found the records for these movements are valid. As mentioned in the previous commentaries, these movements are exceptional with very few shipments during the analysis period. The aerial transport is possible, but we cannot validate this type of information.

Although these movements are exceptional, we decided to keep the records in the analysis to have the highest representativeness of the movements. We also explain this long-distance movement in the manuscript. See the lines 468-470.

Line 439: Please consider rephrasing the loops explanation.

Thank you for your feedback. We have created a more concise and readable version of this paragraph. Please see the lines 415 – 419.

Line 366: Please consider which was the first assumption and based on what analysis.

We appreciate your feedback. The original text was unclear, so we have revised it to make it more concrete. Please see lines 359-364.

Line 404: Please consider separating methods and results in this paragraph.

The fragment of methods was placed in lines 217 – 219, while the results were placed in line 390-391.

Line 420: Please consider rephasing this paragraph.

The paragraph has been revised to enhance clarity and conciseness. The updated version is lines 396-400.

Line 425: Please consider explaining better why on the map there are NA. My understanding is that 11 communities were identified, the largest 6 are plotted on the map and the other 5 (weaker) are plotted as NA.

Thanks for your comment. We had an error in identifying small communities as NA on the map. We updated the map to show only the label of the large communities for improve clarity and interpretation. We also update the caption to inform readers about this issue (line 409-410).

Line 428: Please consider providing a clearer explanation of “geographically clearly defined” when looking at the map (Fig. 5B) as the colours appear quite scattered, particularly the blue ones. Additionally, regarding the “correspondence” of hotspots to the communities: C4 corresponds to the “red” community, but one municipality is on the opposite northeast side of the country. C2 is divided between the “blue”, “green”, “yellow”, NA (hard to tell); C1 is within the blue community, with some NA as well. Maybe a count of the match could help or use “approximate” in reference to the hotspots.

Although there is a geographic overlap between the hotspots and the identified communities, the correlation is not perfectly defined or clear. Therefore, we decided to omit this statement from the manuscript to avoid confusion.

Line 459: Please consider that this paragraph only addresses the industrial perspective, whereas backyard producers might find it easier to slaughter animals on-site. There is evidence of movements to and from markets and maintaining them for ~1o monthly movements seems economically inefficient (please consider include the number of those markets).

We recognize the limitations in describing the movement profile of small producers due to their lack of regularity in reporting their mobilization. The data gives us a good starting point to understand the swine movement patterns in Mexico and could inspire more focused research in the future.

The number of these markets has been included in the correlation analysis with the number of livestock fairs. This helped us to better understand their influence on the pig movement network.

Line 467: Please consider that movements to slaughterhouses are usually considered as “dead-ends” with respect to disease transmission. How the network without dead ends would look like for using in a disease transmission model?

A network without “dead-ends” would involve movements that eventually return to the original location or continue circulating within the network. This type of network could potentially result in increased disease transmission dynamics, as infected animals could reintroduce the pathogen back into the network instead of being removed from it through slaughter.

However, this is not common on farms in Mexico because animals are usually sent to slaughterhouses or points of sale after a certain fattening period. This creates a unidirectional movement that does not include a return to the farm of origin. This argument is further supported by the low percentage of reciprocity observed in the networks (Please see the Table 3).

We made the paragraph more summarized. See the lines 477-482.

Line 476: Please consider rephrasing this paragraph.

In response to the other reviewer's request for a more concise discussion, we have decided to remove this paragraph.

Line 551: Please consider resume this long paragraph.

To be more concrete in the discussion, we resume the paragraph. See the lines 472 – 475.

Reviewer #2: The paper "Analysis of swine movement network in Mexico: A perspective for disease prevention and control" uses social network analysis (SNA) methods to investigate patterns in Mexico's officially recorded swine movement data. The authors provide the analysis results through plots and tables and discuss their implications. While this kind of analysis is not new, the authors claim this is an original application of SNA in Mexico's swine movement networks. Here is my feedback based on what I understood from reading the paper.

1. The analysis is presented at the municipality level (as nodes) and not the farm level. Hence, there are limitations in interpreting the results. These limitations should be clarified to the readers.

We appreciate your feedback. Thank you for your feedback. These limitations have already been addressed in the manuscript. It is clarified that the analysis is presented at the municipality level rather than the farm level, and the implications for interpreting the results are explained. (Please see the lines 110 - 112).

2. The author's estimate of about 95.9% being attributed to slaughterhouse movements could be misleading if it is just based on shipment counts rather than pig population. Heads in fattening shipments are almost four times more than slaughterhouse shipments. This should be clarified in the wording.

We have added the suggested clarification in the manuscript. Please see the lines 270-271.

3. Some of the statements in the paper are too trivial. For example, statements such as "movements occur in the same areas where swine production is concentrated..." do not provide any interesting insight. The authors should limit these statements and make the write-up concise.

Thank you for your feedback. We've made the manuscript more concise and removed trivial statements.

4. Lines 191-193: The authors use and report Euclidean distance to measure distances between municipalities/nodes. However, shipment distances computed from road networks will likely better fit this context.

Thank you for your feedback. However, this approach presents a significant challenge due to the lack of precise farm location data. We use the centroid of municipalities to calculate the Euclidean distance. Consequently, using road networks would not improve the accuracy of our distance estimates. Given the differences in the area of municipalities, it would be difficult to accurately estimate the most appropriate road routes. This could lead to confusion and redundancy, as all trips would originate from the same node and geographic location. In summary, we believe this recommendation is not feasible.

5. The discussion section is too long, where the authors provide a lot of commentary, but for the reader, it is difficult to extract information that could be used in later studies. For example, what are the top 2 or 3 insights/findings that are novel contributions of this paper?

We agree with your opinion and have adjusted the discussion to make it more concise.

In response to your question, we have identified 3 contributions of this paper:

1. This study describes for the first-time swine movement patterns in Mexico, providing valuable information that can strengthen disease control and prevention strategies and improve the effectiveness of swine traceability procedures.

2. Identification of significant nodes and commercial communities: The analysis identified four main hotspots that concentrate the largest pig population in Mexico, revealing a complex flow of pig movements that also have connections with the rest of the municipalities in the country.

3. Our results mainly reflect the large-scale swine industry in Mexico, but more research is needed to understand the dynamics of smallholder swine movements.

6. Lines 262-264: Grammar issues.

We have considered your recommendation and have made the text more concise. Please see lines 259-260.

---

## [Decision Letter · Decision Letter 1]

12 Aug 2024

Analysis of the swine movement network in Mexico: A perspective for disease prevention and control.

PONE-D-23-44053R1

Dear Dr. Rico-Chávez,

We’re pleased to inform you that your manuscript has been judged scientifically suitable for publication and will be formally accepted for publication once it meets all outstanding technical requirements.

Kind regards,

Clement Adebajo Meseko, DVM, PhD

Academic Editor

PLOS ONE

Additional Editor Comments (optional):

Reviewers' comments:

Reviewer's Responses to Questions

**Comments to the Author**

1. If the authors have adequately addressed your comments raised in a previous round of review and you feel that this manuscript is now acceptable for publication, you may indicate that here to bypass the “Comments to the Author” section, enter your conflict of interest statement in the “Confidential to Editor” section, and submit your "Accept" recommendation.

Reviewer #1: All comments have been addressed

Reviewer #2: All comments have been addressed

2. Is the manuscript technically sound, and do the data support the conclusions?

Reviewer #1: Yes

Reviewer #2: Yes

3. Has the statistical analysis been performed appropriately and rigorously? 

Reviewer #1: Yes

Reviewer #2: Yes

4. Have the authors made all data underlying the findings in their manuscript fully available?

Reviewer #1: Yes

Reviewer #2: Yes

5. Is the manuscript presented in an intelligible fashion and written in standard English?

Reviewer #1: Yes

Reviewer #2: Yes

6. Review Comments to the Author

Reviewer #1: The authors have addressed all the recommendations, resulting in a clearer manuscript. Although the number of pages has not been reduced for conciseness, I consider the manuscript sufficiently clear and recommend it for publication.

Reviewer #2: (No Response)

7. PLOS authors have the option to publish the peer review history of their article (what does this mean?). If published, this will include your full peer review and any attached files.

Reviewer #1: **Yes: **Alfredo Acosta

Reviewer #2: No

---

## [Editor Report · Acceptance letter]

22 Aug 2024

PONE-D-23-44053R1 

PLOS ONE

Dear Dr. Rico-Chávez, 

I'm pleased to inform you that your manuscript has been deemed suitable for publication in PLOS ONE. Congratulations! Your manuscript is now being handed over to our production team.

Kind regards, 

on behalf of

Dr. Clement Adebajo Meseko 

Academic Editor

PLOS ONE